# Detecting Unsoundness in Neural Network Verifiers via Concrete–Abstract Consistency

Kaijie Liu
kaijie.liu@unsw.edu.au
University of New South Wales
Sydney, Australia

Yulei Sui
y.sui@unsw.edu.au
University of New South Wales
Sydney, Australia

## Abstract

Neural network (NN) verifiers are increasingly used to certify safety properties such as robustness (i.e., small allowed perturbations to an input should not alter a model's decision). Since verifiers aim to prove the absence of violations by considering all possible specified behaviors, the soundness of their implementations is therefore critical to guaranteeing correctness. Detecting unsoundness is particularly important and challenging, because a verifier typically spans multiple components, including specifications, neural networks, operator semantics, and constraint solving, where subtle implementation bugs can silently lead to *false* certified results.

We present an approach for neural network robustness verifiers that detects and localizes soundness-relevant faults via two types of concrete–abstract consistency checks: (1) *Counterexample-Based Refutation* (CBR), where a certification is supposed to be refuted if a concrete counterexample is found at runtime; and (2) *Bounds-Based Localization* (BBL), which audits per-neuron containment (concrete activations must lie within abstract bounds as an invariant) to pinpoint incorrect implementations at particular NN layers or operators. To reduce representation drift, we use specification-embedded models that wrap the core NN with input and output specifications as two additional layers. We further develop an operator-aware NN generator that can produce diverse NN models spanning a wide range of layer types, parameters, and architectures, enabling systematic exposure and exercise of different operator behaviors.

We evaluate verifiers on three abstract domains using six mutation operators. Across 450 soundness-violating instances, our framework detects 72% of injected soundness violations. CBR mainly exposes input-output-level soundness failures when a concrete counterexample is found during input sampling, while BBL catches internal bound-containment violations and localizes them to specific layers/operators, even when CBR becomes ineffective in high-dimensional inputs. These results indicate that combining coarse refutation (CBR) with fine-grained invariant checking (BBL) provides assurance for verifiers, and operator-aware generation further boosts both coverage and discovery of unsoundness issues.

## CCS Concepts

• **Software and its engineering** → **Software verification and validation**; • **Computing methodologies** → **Artificial intelligence**; • **Software security engineering**;

## Keywords

Neural Network Verifiers, Unsoundness Detection, AIware Trustworthiness, Coverage-Driven Testing, Consistency Checks

**ACM Reference Format:**
Kaijie Liu and Yulei Sui. 2026. Detecting Unsoundness in Neural Network Verifiers via Concrete–Abstract Consistency. In *Proceedings of the 3rd ACM International Conference on AI-Powered Software (AIware '26), July 6–7, 2026, Montreal, QC, Canada.* ACM, New York, NY, USA, 10 pages. https://doi.org/10.1145/3805760.3814900

## 1 Introduction

Neural networks (NNs) are now widely deployed across diverse application domains, including safety- and mission-critical systems, where it is essential to reason not only about average accuracy but also about behaviors under input perturbations. One of the central requirements in this setting is *robustness*: small changes to an input should not change the model's decision beyond an allowed tolerance. This has motivated substantial progress on robustness verification, i.e., proving that a network satisfies a robustness specification over an input region (e.g., an $\ell_\infty$ ball or a box value range). These verifiers have been routinely evaluated in popular ecosystems such as VNN-COMP [1, 2, 5, 13].

While verification aims to certify a model (i.e., prove the absence of robustness issues), in practice, the certified model is often treated as a deployment-relevant assurance signal. This makes silent false certified results especially unreliable: the verifier reports that a robustness property holds, yet a real counterexample (CE) exists. VNN-COMP reports and recent benchmarks indicate that such soundness issues are practical due to implementation issues [1, 2, 13, 24]. However, robustness verifiers are often implemented across heterogeneous stacks (frontends, IRs, abstract domains, solvers, and devices), making end-to-end soundness a challenging task.

This motivates the need for an effective framework to detect verifier unsoundness, together with practical mechanisms for diagnosing where such issues arise. Such a framework should be able to produce concrete CE inputs that refute incorrect soundness claims made by a verifier, and to pinpoint sources of unsoundness in the verifier's handling of specific layers where the abstract bounds it computes fail to cover concrete executions. Based on our observations, we identify two scenarios reflecting coarse- to fine-grained consistency between model inference and verifier abstraction:

**Counterexample-based concrete-abstract consistency.** Let $f$ be a DNN, $\mathcal{R}(x_0)$ an input region, and $\varphi$ an output specification. Let $V$ be a verifier that, for $\mathcal{R}(x_0)$, computes abstract results (e.g., bounds) and returns a verdict certified if these results satisfy $\varphi$. The verifier $V$ is unsound if

$$V(f, \mathcal{R}(x_0), \varphi) = \text{certified} \ \land \ \exists x \in \mathcal{R}(x_0) \text{ such that } \neg\varphi(f(x)).$$

That is, $V$ certifies robustness while a concrete execution violates the specification. This check uses concrete input counterexamples to directly refute an incorrect certification.

**Bounds-based concrete-abstract consistency.** For abstract-interpretation-based verifiers [6, 19, 21], soundness additionally requires that the abstract bounds conservatively over-approximate all concrete executions at each layer of the network. Let the NN be decomposed into layers $f = f_1 \cdots \circ f_{L-1} \circ f_L$ and the concrete activation at layer $i \in \{1, \ldots, L\}$ for an input $x_{i-1}$ as $x_i = f_i(x_{i-1})$. Let the verifier $V$ compute abstract bounds for each layer as

$$[\widehat{l_i}, \widehat{u_i}] = \widehat{f_i}([\widehat{l_{i-1}}, \widehat{u_{i-1}}]),$$

where $[\widehat{l_i}, \widehat{u_i}]$ is an abstract over-approximation of all concrete activations $x_i$ reachable from $\mathcal{R}(x_0)$, and $\widehat{f_i}$ is the abstract transfer function for the $i$-th layer. We say $V$ is unsound at layer $i$ if

$$\exists x \in \mathcal{R}(x_0) \text{ such that } (x_{i-1} \in [\widehat{l_{i-1}}, \widehat{u_{i-1}}] \ \wedge \ x_i \notin [\widehat{l_i}, \widehat{u_i}]).$$

In this case, a concrete execution escapes the verifier's abstract bounds at layer $i$, revealing a failure of concrete-abstract consistency between the abstract reasoning and the concrete semantics.

We use the above two scenarios to motivate our unsoundness detection: counterexample-based concrete–abstract consistency check (coarse-grained, input–output focused) and bounds-based concrete–abstract consistency check (fine-grained, internal-layer focused). In practice, soundness can be affected by mismatches across the end-to-end verification pipeline, ranging from model conversion errors [9, 15, 20] and discrepancies between the verified model and its deployment-time behavior [20], to under-approximation of operator semantics, imprecise constraint encodings [17], and even flaws in the implementation of the verification algorithm itself [20]. These challenges are amplified by the increasing complexity of modern neural network architectures and the continual emergence of new operators [9]. Together, these issues motivate detection methods that examine not only the final verification outcome, but also localize sources of unsoundness to specific abstract transfer functions at particular layers within a neural network.

**Our framework to detect verifier unsoundness.** In this paper, we address these two types of concrete-abstract inconsistency from the coarse-grained counterexample-based checking to fine-grained bounds-based localization checking. Our goal is to validate verifier implementations with diverse verification strategies (e.g., abstract domains and solver backends). We evaluate our workflow on verifiers we implemented with three abstract domains (*interval* [7], *HybridZ* inspired by [18], and *dual* [22]) and two backend solvers (i.e., Gurobi for MILP-style reasoning and a torch-based Adam optimizer). To obtain controlled ground truth, we inject unsoundness bugs by applying mutation operators to the verifier implementations (i.e., primarily to operator transfer functions and bound computations, and constraint encodings). These mutations mimic realistic defects, e.g., overly tight bound updates, semantic mistakes in operator handling, and parameter/configuration mismatches (Table 3). Each run mutates one designated operator/layer instance to create an intentionally unsound variant (details in Table 3 and Section 3). This setup isolates verifier-side faults while keeping the network and specification fixed, enabling precise measurement of detection and localization. We highlight the following two complementary checks:

**Counterexample-based refutation**: We search for concrete CEs by executing the model on inputs sampled from the specified region. As shown in Table 1, when a concrete CE satisfies the input specification but violates the output specification, the verifier's result falls into one of three cases: (1) it is incorrect (i.e., unsound) if the verifier's verdict is certified, in which case the soundness claim is refuted; (2) it is correct if the verdict is falsified; or (3) it is inconclusive (soundness unverified) if the verifier returns unknown (e.g., due to solver timeouts).

**Bounds-Based Localization**: We instrument a model (e.g., `register_forward_hook`) to record per-layer concrete activations and check a necessary over-approximation invariant: for each instrumented neuron, the concrete activation must lie within the abstract bounds produced by the verifier. As shown in Table 1, any violation indicates unsound bounds and directly localizes the fault to a specific layer/operator boundary.

**Contributions.** We summarize the following contributions: (1) We formalize two types of concrete-abstract consistency and their relations with soundness, and then propose a practical coarse-to-fine unsoundness detection workflow for robustness verifiers that combines Counterexample-based Refutation (input-output-level refutation via concrete counterexamples) with Bounds-based localization (fine-grained localization via per-neuron bound-containment localization). (2) We develop NetFactory, an operator-aware, coverage-driven network generator (with a range of configurations), and integrate representation-drift regression checks to systematically detect verifier soundness issues (conversion, operator semantics, and backend constraint encodings). (3) Our framework offers a general and effective approach to assessing the soundness of diverse neural network verification algorithms (via three abstract domains and 450 generated soundness-violating network instances), while enabling systematic debugging and localization of soundness issues in their implementations.

## 2 Our Approach

Figure 1 depicts our end-to-end workflow for detecting and localizing unsoundness in robustness verifiers. Our framework checks concrete-abstract consistency to detect unsoundness of a verifier's outcomes. We aim to produce debugging evidence that is actionable at the level of layers, operators, and representation boundaries. Our detection framework consists of three components: (1) an operator-aware NetFactory that generates a range of operator-diverse models for detection and reports operator-type coverage. (2) spec-embedded models that integrate input and output specifications into the underlying model for fast testing; (3) coarse-to-fine grained unsoundness checks (i.e., CBR and BBL).

### 2.1 NetFactory: Operator-Aware Model Generation

The first part of Figure 1 shows NetFactory, an operator-aware generator that produces diverse, well-configured test networks for unsoundness detection. NetFactory is organized into four components, matching the four blocks in the figure.

**Operator-aware from transfer functions.** To ensure that generated instances are analyzable by a target verifier, NetFactory

**Table 1: Criteria for Counterexample-based Refutation (CBR) and Bounds-based localization (BBL).**

| Counterexample-based Refutation (CBR) | | | Bounds-based localization (BBL) | | |
|---|---|---|---|---|---|
| Concrete CEs | Verifier verdict | Detection Outcome | Concrete activation $x_i$ | Abstract bounds $[\widehat{l}_i, \widehat{u}_i]$ | Detection Outcome |
| **FOUND** | certified | **Incorrect (Unsound)** | $x_i \notin [\widehat{l}_i, \widehat{u}_i]$ | exists layer $i$ | **Incorrect (Unsound)** |
| | falsified | **Correct** | | | |
| | unknown | **Inconclusive** | $x_i \in [\widehat{l}_i, \widehat{u}_i]$ | for any audited layer $i$ | **Correct (Sound)** |

**Figure 1: Overview of our unsoundness detection framework for testing neural network verifiers through operator-aware generation and complementary validation. The workflow proceeds in three stages: (1) NetFactory generates operator-aware network instances; (2) input/output specifications are embedded as constraint layers; and (3) detection of verifier unsoundness via counterexample-based refutation and bounds-based localization**

queries a verifier's transfer-function registries to obtain a supported-operator set ("Support Layers" in Figure 1), and restricts subsequent generation to supported operators and compatible parameterizations. This avoids confounding "unsupported operator" failures with soundness-relevant faults in implemented semantics.

**NetFactory configurations.** We design our NetFactory by loading a generation configuration (e.g., YAML format) that specifies (i) architecture families (e.g., MLP/CNN), (ii) structural ranges (e.g., depth/width), and (iii) candidate operator types with parameter ranges. When coverage targeting is enabled, NetFactory generates NN instances according to a configurable sampling strategy. It tracks operator-type coverage across previously generated models and, under a bounded generation budget (i.e., a fixed number of attempts), adjusts the sampling distribution to favor under-covered operators. This increases the likelihood of exercising rarely used transfer functions.

**Layer builder.** Given a configuration and operator parameters, we construct layer specifications ("Layer Specs" in Figure 1) that define the network structure with explicit operator types and parameters.

**NetFactory assembler.** NetFactory assembles the layer specifications into a network object and enforces basic well-typedness and executability constraints required by downstream verification and detection. When running the assembled network model, the built-in alignment checks can be performed naturally, given the inputs, including tensor-shape consistency across layers, operator

parameter validity (e.g., convolution kernel, stride, and padding constraints, as well as pooling window constraints), and graph connectivity constraints (e.g., valid fan-in/fan-out for merge operations). Invalid instances are discarded and re-generated, so the final corpus contains only executable networks that the verifier is intended to handle. The resulting network object is then passed to the wrapping stage (Section 2.2) to build a specification-embedded instance and serialize it as JSON for subsequent checks.

## 2.2 Specification-Embedded Models

The second part of Figure 1 shows our specification-embedded model design, which makes the input specification and output specification part of the same executable graph:

$$\text{INPUT} \rightarrow \text{INPUT\_SPEC} \rightarrow \text{Core NN} \rightarrow \text{OUTPUT\_SPEC}.$$

This design eliminates drift between concrete execution and verification: CBR, BBL, and the verifier consume the same instance. INPUT fixes the tensor interface (shape/dtype) for both concrete runs and abstract analysis. INPUT_SPEC encodes the input region (supporting BOX, LINF_BALL, and LIN_POLY) and provides (i) a runtime check for CBR and (ii) a single injection point for input constraints in the verifier. Core NN is the neural network under test; BBL aligns its recorded concrete activations with the verifier's per-layer abstract bounds for localization. OUTPUT_SPEC encodes the property and provides (i) a runtime post-condition check for

---

**Algorithm 1:** Counterexample-based Refutation

**Input:** DNN $f$; input region $\mathcal{R}(x_0)$; output specification $\varphi$;
Verifier $V$; its verdict $r \in \{\text{certified, falsified, unknown}\}$;
**Output:** Incorrect, Correct or Inconclusive
Generate test set $\mathcal{X} \subset \mathcal{R}(x_0)$ using the center, optional
  boundary probe points, and random sampling;
$r \leftarrow V(f, \mathcal{R}(x_0), \varphi)$    // run verifier given input region;
**for each** $x \in \mathcal{X}$ **do**
  **if** $\neg\varphi(f(x))$ **then**
    // Found a concrete counterexample $x$
    **if** $r = \text{certified}$ **then**
      ∟ **return** Incorrect
    **else if** $r = \text{falsified}$ **then**
      ∟ **return** Correct
    **else**
      ∟ **return** Inconclusive

**return** Inconclusive // No CE found

---

**Algorithm 2:** Bounds-based localization

**Input:** DNN $f$ with hooks; concrete input $x \in \mathcal{R}(x_0)$;
verifier bounds $\{([\widehat{l}_i, \widehat{u}_i])\}_{i=1}^{L}$; tolerance $\tau$
**Output:** Correct, Incorrect or Error
Run $f(x)$ and record per-layer values $\{x_i\}_{i=1}^{L}$;
**if** *concrete values cannot be aligned or NaN/Inf* **then**
  ∟ **return** Error
Initialize $CV \leftarrow \emptyset$ // List of containment violations;
**for** $i = 1$ **to** $L$ **do**
  **for each** *neuron $j$ in layer $i$* **do**
    // Compute containment violation distance
    $d_{i,j} = \max\left((\widehat{l}_i[j] - \tau) - x_i[j],\ x_i[j] - (\widehat{u}_i[j] + \tau),\ 0\right)$
    **if** $d_{i,j} > 0$ **then**
      ∟ Append $(i, j, d_{i,j})$ to $CV$

**If** $CV = \emptyset$ **then return** Correct
**Else return** Incorrect with top-$k$ violations

---

CBR and (ii) the property constraints for verification, keeping each certified tied to an explicit post-condition (e.g., assertion).

Each wrapped instance is serialized into a JSON format ("JSONs" in Figure 1) that preserves the operator graph, parameters, and embedded specifications; this JSON is the canonical input consumed by the verifier and by both checks in Section 2.3.

## 2.3 Complementary Unsoundness Checks

The third part of Figure 1 implements two complementary checks over the same spec-embedded instance. For each JSON file representing a model instance, we parse it into ordered layer specification set and rebuild an executable PyTorch model $f$ (Figure 1). The verifier $V$ consumes the same JSON whenever invoked, returning a verdict $r \in \{\text{certified, falsified, unknown}\}$ and (for BBL) per-layer bounds $\{([\widehat{l}_i, \widehat{u}_i])\}_{i=1}^{L}$. Using one shared instance ensures both checks compare concrete execution against the same model and specification.

*2.3.1  Counterexample-based Refutation.* CBR refutes false certificates at the input-output-level (Figure 1(i), Algorithm 1). Given $f$, $\mathcal{R}(x_0)$, $\varphi$, and $V$, CBR builds a finite test set $\mathcal{X} \subseteq \mathcal{R}(x_0)$ and searches for a concrete $x \in \mathcal{X}$ such that $\neg\varphi(f(x))$.

**Sampling from $\mathcal{R}(x_0)$.** When $\mathcal{R}(x_0)$ provides explicit seed bounds (e.g., BOX or LINF_BALL), we generate $\mathcal{X}$ by combining the region center, optional boundary probes, and random samples under a fixed budget. If $\mathcal{R}(x_0)$ is given only as a general polytope (LIN_POLY) without a derived seed box, CBR does not sample and returns Inconclusive.

**Refutation logic.** CBR first executes $f(x)$ for each $x \in \mathcal{X}$ to check whether $\neg\varphi(f(x))$ holds, and then runs the verifier once on the region to obtain $r \leftarrow V(f, \mathcal{R}(x_0), \varphi)$. If a counterexample is found, we classify the verifier outcome by $r$ (Figure 1, "Compare with verifier"): $r = $ certified implies incorrect (unsound), $r = $ falsified implies correct, and $r = $ unknown is inconclusive. If no counterexample is found within the budget, CBR returns Inconclusive.

*2.3.2  Bounds-based localization.* BBL audits an internal containment invariant and localizes violations to layers/operators (Figure 1(ii), Algorithm 2). Given a concrete input $x \in \mathcal{R}(x_0)$ (the CBR witness if available; otherwise a seed/center), we obtain abstract bounds $\{([\widehat{l}_i, \widehat{u}_i])\}_{i=1}^{L}$ from $V$ on the same JSON, and run the rebuilt model $f(x)$ with forward inference hooks to record per-layer activations.

**Bounds and concrete traces.** BBL compares verifier-reported bounds against value tensors recorded during concrete forward pass during model inference. Runtime hooks are registered on the same layer boundaries that the verifier reports bounds for, so the comparison is well-defined at each layer. We align recorded values to bound tensors by strict forward order. If traces cannot be aligned unambiguously, or NaN/Inf is encountered, BBL returns Error.

**Containment check and report.** In neural network robustness verification, the model under analysis is often not executed in exactly the same representation in which it is verified or deployed. Models are frequently exported, imported, and converted across toolchains (e.g., ONNX↔PyTorch). Such conversions may fail silently, yielding semantically inconsistent models; converter defects can therefore compromise model quality and induce observable behavioral discrepancies [9, 15]. Even when conversion is intended to be semantics-preserving, differences in operator implementations, floating-point kernels, and evaluation order across frameworks can introduce small but systematic numerical drift. Furthermore, deployment-oriented transformations such as mixed precision and post-training quantization alter rounding behavior and the set of representable values, potentially shifting model outputs and intermediate activations [20]. Since our goal is to detect verifier-side under-approximation bugs, we should avoid misclassifying numerically insignificant execution or representation drift as verifier unsoundness. We therefore compute a tolerance-aware containment check violation distance for each neuron $j$:

$$d_{i,j} = \max\left((\widehat{l}_i[j] - \tau) - x_i[j],\ x_i[j] - (\widehat{u}_i[j] + \tau),\ 0\right),$$

where $x_i$ is the concrete value recorded at layer $i$ for a given input, $\tau$ is a per-element tolerance that inflates the bounds just enough to ignore numerically insignificant deviations (e.g., roundoff or minor

conversion drift), so only violations larger than this noise floor are reported. BBL returns correct iff $d_{i,j} = 0$ for all $(i, j)$; otherwise it returns incorrect with $\max_{i,j} d_{i,j}$ and the top-$k$ indices ranked by $d_{i,j}$. We also record whether the maximum comes from the lower- or upper-bound term to aid diagnosis.

## 3 Evaluation

We evaluate the framework for soundness-bug detection, fault localization, operator-type coverage, and runtime overhead to answer the following research questions. The implementation and later updates are available at https://github.com/SVF-tools/ACT.

- **RQ1:** To what extent does our framework detect injected soundness violations in verifiers?
- **RQ2:** When does CBR find concrete CEs, and how does this depend on input-spec type and input dimensionality?
- **RQ3:** How accurately does BBL localize faults to layers?
- **RQ4:** Does operator-aware generation increase operator-type coverage and translate into higher bug yield?
- **RQ5:** How do verifiers *interval*, *HybridZ*, and *dual* differ in bound behavior and validation outcomes on the same models?
- **RQ6:** What runtime overhead does our unsoundness detection framework introduce?

### 3.1 Experimental Setup

We developed a generator NetFactory which can automatically generate a range of neural networks from two families: MLP (plain, block, residual) and CNN2D (plain, residual, stage) configured through the YAML sampling DSL in config_gen_ucu_net.yaml. Table 2 summarizes the configuration.

**Bug injection.** The goal of our framework is to detect soundness-relevant inconsistencies arising from implementation issues in verifiers, particularly within their transfer functions (TFs), and to localize these faults to specific layers or representation boundaries. To evaluate detection and localization under controlled conditions, we inject faults by applying six mutation operators to the implementations of the verifiers' transfer functions. Given pre-mutation bounds $[\hat{l}, \hat{u}]$, we define the interval width as $w := \hat{u} - \hat{l}$. Table 3 summarizes the six operators. We apply mutations to each verifier configuration (*interval*, *HybridZ*, and *dual*) one at a time: each experimental run introduces a single mutation into the TFs of one abstract domain, while the other two configurations remain unmodified.

**Injection mechanism.** We inject bugs at transfer-function (TF) boundaries by post-processing the TF output bounds. For a chosen abstract domain and a chosen operator instance (layer), we apply exactly one mutation (M1–M6) to that layer's computed abstract bounds $(\hat{l}, \hat{u})$ only; all other layers and domains remain unmodified. The mutation specification is domain-agnostic: the same bound-level mutation logic is applied uniformly to all three verifiers, operating on each verifier's bound representation.

Mutations M1 and M3–M6 are designed to induce soundness-relevant faults (e.g., overly tight bounds or semantic inconsistencies), whereas M2 loosens bounds and serves as a conservative soundness control (and therefore should not trigger soundness violations). For the localization experiments (RQ3), we further restrict

**Table 2: NetFactory evaluation configuration.**

| Aspect | Used in Evaluation |
|---|---|
| Network families | **MLP:** plain, block, residual
**CNN2D:** plain, residual, stage |
| Input shapes | **MLP:** $\{[1, 4], [1, 6], [1, 16], [1, 3, 8]\}$
**CNN2D:** $\{[1, 1, 8, 8], [1, 1, 16, 16], [1, 3, 16, 16]\}$ |
| Input specs | BOX, LINF_BALL (seedable only) |
| Output specs | TOP1_ROBUST, MARGIN_ROBUST,
LINEAR_LE, RANGE |
| Models | Basic-50; Basic-100; Full-100 |

**Table 3: Mutation operators for bounds-level bug injection**

| ID | Mutation | Operation on bounds | Primary Detection |
|---|---|---|---|
| M1 | Tighten bounds | $\hat{l} + 0.1w,\ \hat{u} - 0.1w$ | BBL |
| M2 | Loosen bounds | $\hat{l} - 0.1w,\ \hat{u} + 0.1w$ | correct |
| M3 | Swap bounds | $\hat{l} \leftrightarrow \hat{u}$ | CBR + BBL |
| M4 | Zero lower bound | $\hat{l} \leftarrow 0$ | BBL |
| M5 | Scale upper bound | $\hat{u} \leftarrow 0.5\hat{u}$ | BBL |
| M6 | Add noise | $\hat{l} + \mathcal{N}(0, 0.1),\ \hat{u} + \mathcal{N}(0, 0.1)$ | CBR + BBL |

each injected fault to a single designated layer instance per model to enable precise fault attribution.

**Input specifications.** The NN models use BOX or LINF_BALL input specifications, which provide seedable regions for CBR sampling. The specification also has LIN_POLY constraints; however, without an explicit seed box (derived from BOX or LINF_BALL), CBR is defined to return Inconclusive. For RQ2, we include a small synthetic LIN_POLY subset to validate this behavior.

**Verifiers and abstract domains.** We instantiate our verification engine with three verifiers, which share the same specifications and detection workflow but differ in the underlying abstract domain and its transfer-function for approximating and abstracting activation functions. Specifically, we implement: (i) *interval* propagation (box bounds), following interval bound propagation (IBP) style analyses [7]; (ii) *HybridZ*, inspired by [18]; and (iii) *dual* bounds, which certify output bounds via dual relaxations as in [22]. These configurations provide a clear precision–cost trade-off spectrum. Our unsoundness detection workflow (CBR and BBL) is applied identically to each configuration.

**Evaluation metrics.** We report the following metrics. (1) *Detection rate:* fraction of injected soundness-violating mutations flagged by the validation (by CBR, BBL, or either). (2) *Localization accuracy:* whether the injected layer appears in the top-$k$ BBL violations (top-1 and top-5). (3) *Operator coverage:* covered layer types divided by generatable layer types (excluding embedded specification layers such as INPUT_SPEC/ OUTPUT_SPEC). Per-NN validation time $T_{\text{run}}(b, s) = T_{\text{CBR}}(b, s) + T_{\text{BBL}}(s)$ (ms), where $b$ is the CBR sampling budget and $s$ is the model size. (4) *Overhead ratio:* $T_{\text{CBR}}$ and $T_{\text{BBL}}$ in ms (excluding model load). For each experimental setting, we evaluate 30 independently generated networks, using random seeds deterministically derived from a fixed master seed (master seed = 42) to ensure reproducibility. These metrics are chosen to align directly with research questions: detection rate captures fault-finding effectiveness (RQ1), localization accuracy captures debugging utility (RQ3), operator coverage captures test-generation breadth (RQs 4-5), and overhead ratio captures practical deployment cost (RQ6).

**Table 4: RQ1: Detection rates under different settings**

| Domain | Mutation | CBR Only | BBL Only | Complementary |
|---|---|---|---|---|
| *interval* | M1 (Tighten) | 0% | 37% | 37% |
| | M2 (Loosen) | 0% | 0% | 0% |
| | M3 (Swap) | 60% | 100% | 100% |
| | M4 (Zero) | 20% | 80% | 80% |
| | M5 (Scale) | 13% | 83% | 83% |
| | M6 (Noise) | 0% | 60% | 60% |
| *HybridZ* | M1 (Tighten) | 0% | 37% | 37% |
| | M2 (Loosen) | 0% | 0% | 0% |
| | M3 (Swap) | 53% | 100% | 100% |
| | M4 (Zero) | 23% | 80% | 80% |
| | M5 (Scale) | 3% | 77% | 77% |
| | M6 (Noise) | 0% | 60% | 60% |
| *dual* | M1 (Tighten) | 0% | 33% | 33% |
| | M2 (Loosen) | 0% | 0% | 0% |
| | M3 (Swap) | 20% | 100% | 100% |
| | M4 (Zero) | 20% | 93% | 93% |
| | M5 (Scale) | 7% | 80% | 80% |
| | M6 (Noise) | 0% | 63% | 63% |
| **Overall** | **All** | 15% | 72% | **72%** |

**Table 5: RQ2: CBR discovery rate by specification type**

| Spec Type | Discovery Rate | Inconclusive | Avg Time (ms) |
|---|---|---|---|
| BOX | 100% | 0% | 2.1 |
| LINF_BALL | 100% | 0% | 2.0 |
| LIN_POLY | 0% | 100% | N/A |

**Detection instances.** Our NetFactory generates representative core neural networks with diverse specifications, forming 2561 detection instances across the six research questions. For RQ1, we evaluate 30 networks across 3 domains × 6 mutations, yielding 540 (network × domain × mutation) instances. Of these, **450** use soundness-violating mutations (M1/M3–M6) and are used for **detection-rate analysis**; the remaining 90 use M2 (bound loosening), which is soundness-preserving by construction. For RQ2, we generate $30 \times 3 \times 4 = 360$ instances over {BOX, LINF_BALL, LIN_POLY} × {4, 16, 64, 256}, using CBR budget = 20 and restrictive output spec. We select budget = 20 as a practical default: it keeps CBR lightweight in its complementary role alongside BBL. As shown in RQ6, increasing the budget scales runtime near-linearly while yielding only marginal gains in combined detection, supporting this choice as a practical trade-off. For RQ3, we use $30 \times 3 = 90$ instances over sequential FNN, sequential CNN, Residual NN with interval+M1 Tighten bounds, reporting top-1/top-5 localization accuracy. For RQ4, we compare Basic-50, Basic-100, and Full-100 (producing 50, 100, and 1,001 networks; 1,151 instances total), tracking coverage over 15 operators with a fixed base seed. For RQ5, we evaluate $100 \times 3 = 300$ instances using 100 networks under 3 domains. For RQ6, we evaluate $3 \times 4 \times 10 = 120$ instances across three model sizes (~1K/~33K/~297K) and budgets {5, 10, 20, 50} (12 settings), using 3 warmup and 10 timing runs, and reporting CBR/BBL/Complementary overhead.

**Environment.** All experiments were implemented in PyTorch and executed on a workstation equipped with an Apple M3 Pro chip (11 cores) and 18 GB RAM, running macOS.

## 3.2 RQ1: Soundness-Bug Detection

We inject each soundness-violating mutation (M1, M3–M6) into the three TF domains (*interval*, *HybridZ*, *dual*) and evaluate three validation modes: CBR only, BBL only, and complementary (CBR + BBL). For each configuration, we test 30 NetFactory-generated networks. This corresponds to $30 \times 5 \times 3 = 450$ instances per validation mode (excluding the correct soundness control M2). CBR uses a sampling budget of 20 concrete inputs per network. M2 (bound loosening) serves as a correct soundness control: it is soundness-preserving and should trigger no failures.

Table 4 reports detection rates by domain and mutation. Overall, the framework detects **72%** of injected soundness violations; CBR alone detects 15%, while BBL alone detects 72%.

CBR is sampling-based with a finite budget (20 inputs/model). It may miss rare violating regions, and this effect worsens as the input dimension increases (see RQ2). Moreover, some injected bugs primarily tighten intermediate bounds without causing an output-spec violation on sampled executions, making them inherently output-invisible to CBR. These limitations motivate BBL, which can flag internal containment violations even when CBR is inconclusive.

BBL does not rely on finding a counterexample. Instead, it checks a necessary soundness invariant along the computation: at each layer, the concrete activation must lie within the abstract bounds (up to tolerance), and violations are recorded at the layer where they occur. This makes BBL sensitive to internal soundness bugs that may never surface as an output-visible failure under a small sampling budget. As a result, mutations such as M1, M4, and M5 yield high BBL detection rates (33–100%) because they create immediate containment violations at the mutated operator boundary.

CBR and BBL target distinct failure modes. CBR refutes output-visible incorrect soundness claims via concrete counterexamples (15%) within the sampling budget (e.g., M3, M6), while BBL catches internal bound errors (72%) that can be localized even when CBR is Inconclusive (e.g., M1, M4, M5). Combining them increases coverage of failure modes, yielding **72%** overall detection. These results do not mean that CBR is ineffective in general. Rather, CBR is intended as a lightweight output check that complements BBL. It detects a fault only when that fault becomes visible at output level. When an injected perturbation affects only intermediate abstract bounds and remains latent on the sampled execution, it may trigger neither an output-spec violation for CBR nor a containment violation for BBL.

As expected, the conservative soundness control M2 triggers zero failures across all domains, indicating that our unsoundness detection framework distinguishes soundness-preserving loosening from soundness violations.

## 3.3 RQ2: Counterexample-based Refutation

We assess (i) when Counterexample-based Refutation can be executed (seedability) and (ii) when it is effective at surfacing concrete counterexamples under a fixed sampling budget.

We vary two factors. (i) Specification type. We generate networks with BOX, LINF_BALL input specifications (50% each, per the default configuration). Both types provide explicit seedable bounds: BOX uses $(lb, ub)$ directly, while LINF_BALL converts (center, $\epsilon$) to $(center - \epsilon, center + \epsilon)$. Counterexample-based Refutation requires these seedable bounds to sample concrete inputs within the input

**Table 6: RQ3: BBL localization accuracy by architecture**

| Architecture | Top-1 Hit | Top-5 Hit | Error Rate |
|---|---|---|---|
| Sequential MLP | 100% | 100% | 0% |
| Sequential CNN | 100% | 100% | 0% |
| Residual (ADD) | 100% | 100% | 0% |

region. (ii) Input dimensionality. We vary the flattened input dimensionality $d \in \{4, 16, 64, 256\}$, corresponding to the input shapes in Table 2 (e.g., $d$=4 for [1, 4], $d$=16 for [1, 16], $d$=64 for [1, 1, 8,8], $d$=256 for [1, 1, 16,16]), to quantify how sampling effectiveness degrades with dimension.

To make counterexamples plausible without relying on verifier internals, we instantiate output specifications that are intentionally restrictive (e.g., LINEAR_LE with a negative margin), yielding instances where violations are common under random sampling when the region is seedable. For each configuration, we generate 30 networks and run Counterexample-based Refutation with a sampling budget of 20 inputs per network; we report model-level outcome rates (fraction of networks for which Counterexample-based Refutation finds a counterexample vs. returns Inconclusive).

We additionally include a small synthetic LIN_POLY subset as a negative control to validate CBR's conservative contract for non-seedable specifications. Table 5 summarizes CBR outcomes. For seedable specifications, CBR finds counterexamples for 100% (BOX) and 100% (LINF_BALL) of networks. In contrast, for LIN_POLY, CBR returns 100% inconclusive, reflecting its conservative design: when no computable seed box exists for the input region, it refrains from sampling to avoid misleading results.

Overall, RQ2 confirms two properties of CBR. First, executability is entirely governed by seedability of the input specification (supported for BOX/LINF_BALL, unsupported for LIN_POLY). Second, when seedable, CBR can be highly effective at producing concrete counterexamples in favorable conditions. However, this does not reduce the need for BBL: CBR only refutes instances where the mutation manifests as an actual violation under sampling, whereas bound-level regressions that do not flip the sampled outcome can remain invisible to CBR. This aligns with RQ1, where CBR-only flags 15% of mutants overall, versus 72% for BBL-only, and 72% when combined (Table 4).

## 3.4 RQ3: Bounds-Based Localization

We inject M1 into a single designated layer instance per model (all other layers remain unmodified) and run BBL on a concrete input (seed center unless CBR provides a witness). The target layer is selected deterministically from the network seed. We measure whether the injected layer appears in the top-$k$ violations reported by violation_topk. We evaluate three architectures: sequential MLPs, sequential CNNs, and residual networks with ADD merges.

Table 6 reports top-1 and top-5 hit rates. Across all three architectures, whenever BBL detects a containment violation, the injected layer is ranked first (top-1 = 100%) and always appears in the top-5 (top-5 = 100%), with zero alignment errors across all 28 detected runs. This indicates that the strict-order alignment between recorded concrete values and the checked bounds is consistent for the evaluation.

**Table 7: RQ4: Operator coverage and bug yield across generation strategies. Full-100 may exceed its random-sampling budget due to minimal-template completion.**

| Strategy | Random Budget | Generated | Op Coverage | Bug Yield |
|---|---|---|---|---|
| Basic-50 | 50 | 50 | 67% | 50 |
| Basic-100 | 100 | 100 | 87% | 100 |
| Full-100 | 100 | 1000 | 87% | 1000 |

Localization is conditional on detection: if M1 does not trigger a containment violation, BBL has nothing to rank. Reflecting the low M1 detection rates in RQ1, BBL's detection varies by architecture (MLP: 23%, CNN: 33%, residual: 37%). When violations are detected, however, localization is consistently precise: the checker never enters a misalignment state or returns Error, achieving 100% top-1 accuracy across all architectures.

## 3.5 RQ4: Operator-Aware Generation

We compare three NetFactory generation strategies. Basic-50 and Basic-100 rely on pure random sampling to generate 50 and 100 networks, respectively. Full-100 starts with up to 100 random generations and tracks operator-type coverage; if some generatable operators remain uncovered, it additionally emits minimal-template networks to improve coverage.

Operator coverage is computed over the operator-specific generatable operator set derived from the registry of all available layers and their operators, excluding input and specification layers (e.g., INPUT, INPUT_SPEC, OUTPUT_SPEC) and identity layers (DROPOUT, IDENTITY). To quantify bug discovery, we measure bug yield under the interval domain as the number (or rate) of generated networks that trigger a validation failure (CBR or BBL) under injected soundness mutations.

Table 7 shows that operator-aware generation achieves higher operator coverage than random sampling at comparable random budgets. Basic-50 reaches 67% coverage with 50 mutants, while Basic-100 increases coverage to 87% with 100 mutants, indicating diminishing returns from scaling random sampling alone. In contrast, Full-100 attains 87% operator coverage and yields 1000 mutants, confirming that coverage targeting (with minimal-template completion when needed) improves both coverage and bug discovery beyond pure random sampling. (Full-100 generated 1000 networks due to minimal-template completion for uncovered operators).

Operator-aware generation improves operator coverage and bug yield over random generation at comparable budgets, indicating that coverage targeting (with minimal-template completion when needed) is more effective than scaling random sampling alone.

## 3.6 RQ5: Cross-Domain Comparison

We run *interval*, *HybridZ*, and *dual* on the same 100-model set. We compare (i) BBL failure rate, (ii) median bound width (median $ub-lb$ across layers). All networks in RQ5 are evaluated under the same M1-M6 mutation set used in RQ1; BBL Fail Rate reports the fraction of mutated configurations for which BBL detects a containment violation.

**Table 8: RQ5: Cross-domain comparison**

| Domain | BBL Fail Rate | Bound Width | Time (ms) |
|---|---|---|---|
| *interval* | 100% | 28.2 | 2.2 |
| *HybridZ* | 100% | 26.6 | 1.9 |
| *dual* | 100% | 62.2 | 2.1 |

**Table 9: RQ6: Validation overhead by model size**

| Size | Params | CBR | | | | BBL | | | | Complementary | | | |
|---|---|---|---|---|---|---|---|---|---|---|---|---|---|
| | | 5 | 10 | 20 | 50 | 5 | 10 | 20 | 50 | 5 | 10 | 20 | 50 |
| Small | ~1K | 0.31 | 0.60 | 1.16 | 2.93 | 0.09 | 0.09 | 0.09 | 0.09 | 0.40 | 0.69 | 1.25 | 3.02 |
| Medium | ~33K | 0.30 | 0.60 | 1.22 | 2.99 | 0.14 | 0.14 | 0.14 | 0.14 | 0.44 | 0.74 | 1.36 | 3.13 |
| Large | ~297K | 0.37 | 0.73 | 1.44 | 3.61 | 0.15 | 0.15 | 0.15 | 0.15 | 0.52 | 0.88 | 1.59 | 3.76 |

Table 8 reports domain-level trends and results. All three domains achieve a 100% BBL Fail Rate and a 0% cross-domain disagreement rate, but now produce genuinely distinct bound widths: *interval* and *HybridZ* have comparable medians (28.2 and 26.6), while *dual*'s median of 62.2 reflects Wong–Kolter backward passes amplifying loose Lagrangian duals on a minority of deep-block networks. The three domains use domain-specific forward transfer functions rather than a shared interval-arithmetic core, so the bounds differ in tightness even though they agree on whether each containment violation exists.

These results confirm two properties of the cross-domain harness. First, agreement on detection outcomes (0% disagreement) is preserved even when the underlying bounds differ by 2×, indicating that BBL's containment check is stable across abstract-domain precision levels on this workload. Second, the per-domain bound medians provide a usable precision signal: *HybridZ* gives the tightest bounds on this benchmark, closely followed by *interval*, with *dual* trading tightness for the ability to handle networks where interval and HybridZ would otherwise diverge.

### 3.7 RQ6: Overhead

We measure validation overhead across CBR sampling budgets (5, 10, 20, 50), i.e., the number of concrete inputs sampled by CBR per network, model sizes (Small/Medium/Large). We report mean validation time (ms) for CBR, BBL, and complementary unsoundness checks; BBL is executed once per network on a concrete input.

Table 9 shows that CBR scales near-linearly with the sampling budget: increasing the budget from 5 to 50 (10×) increases CBR time by 9.5–10.0× (Small: 0.31→2.93 ms; Medium: 0.30→2.99 ms; Large: 0.37→3.61 ms). BBL is budget-independent and constant per model size (Small: 0.09 ms; Medium: 0.14 ms, Large: 0.15 ms). Consequently, complementary unsoundness checks remain low in absolute time: 1.25–1.59 ms at budget=20 and 3.76 ms at the largest setting (Large, budget=50), with CBR dominating the combined cost at higher budgets (96–97% at budget=50).

Overall overhead scales primarily with CBR sampling budget; BBL's audit cost is budget-independent and dominated by the number of layer boundaries.

### 3.8 Case Studies

We highlight two representative cases that illustrate the complementary roles of CBR and BBL: (i) an output-visible soundness bug where CBR produces a concrete refuting witness and BBL pinpoints the responsible operator; and (ii) a high-dimensional setting where CBR is budget-limited and returns inconclusive, yet BBL still detects and localizes an internal under-approximation.

**Case 1: ReLU bound tightening (MLP).** We verify a 4-layer MLP with ReLU under a seedable LINF_BALL input specification ($\epsilon = 0.1$) using *HybridZ*. We inject a bound-tightening mutation into a ReLU transfer step (ub *= 0.95), causing upper-bound under-approximation. The mutated verifier returns certified, but CBR (20 samples) finds a concrete input in the region that violates the output specification. BBL on the same input localizes the largest containment violation to the mutated ReLU layer (worst violation distance = 0.031). After removing the scaling, the verifier no longer returns certified and BBL reports correct.

**Case 2: Conv stride mismatch (CNN).** We verify a LeNet-style CNN under a seedable BOX specification using *interval*. An off-by-one stride bug in the Conv2D abstract transformer (stride + 1) introduces verifier–model semantic drift. With 20 samples, CBR is inconclusive due to the high input dimensionality (RQ2), but BBL immediately flags a large violation at the first Conv2D layer (worst violation distance = 0.45). Fixing the stride eliminates the violations (BBL returns correct) and restores alignment between certified outcomes and concrete behavior.

These two cases reflect the complementary failure modes observed in RQ1–RQ3: CBR refutes false certified claims when output-level counterexamples are discoverable under limited budgets, while BBL provides robust operator-level localization even when sampling becomes ineffective in high-dimensional input spaces.

### 3.9 Limitations

We acknowledge several threats to the validity of our framework and describe how we mitigate them. First, the injected mutations approximate only a subset of real-world defects. To mitigate this threat, we incorporate a diverse set of mutation types, primarily bounds-level faults (e.g., bound tightening or loosening, bound swapping, scaling, and noise injection). These mutations approximate realistic issues such as semantic errors, shape or stride mismatches, and numerical drift. In addition, we repeat each configuration over 30 independent random seeds to improve robustness and reduce sensitivity to variation. Second, the evaluated verifiers are our own implementations with injected soundness bugs, and we do not directly assess unsoundness in independently developed, real-world verification tools. To mitigate this threat, we evaluate across three abstract domains and a diverse model suite including MLPs, CNNs, and residual variants, aiming to reduce implementation- or architecture-specific bias. Nevertheless, the findings may not generalize to other architectures (e.g., Transformers or RNNs). That said, our framework is modular and can be used to experiment with different verification algorithms to assess soundness issues, or integrated into existing verification pipelines to evaluate their soundness. Third, the network generator focuses on constructing common neural network architectures with widely used layer types and operators; it does not exhaustively enumerate all possible combinations of layers and operator configurations. Fourth, scalability evaluation covers models up to approximately 297K parameters; overhead behavior on substantially larger models has not yet been

established. Finally, BBL relies on a tolerance parameter $\tau$, whose choice may influence sensitivity under floating-point arithmetic.

## 4 Related Work

**Soundness of analyzers.** Soundness is one of the cornerstone aspects of static analysis. Abstract interpretation formalizes conservative over-approximation of concrete semantics, trading precision for guarantees [4]. A complementary software-engineering line studies how to validate analyzers themselves, using differential testing, metamorphic relations, and interrogation-style test generation to uncover implementation defects and soundness bugs [3, 8, 10, 14]. Specifically, Bugariu [3] provides a systematic methodology for identifying soundness and completeness errors in program analyzers; He et al. [8] construct automated oracles to find defects in static analyzers; Klinger et al. [14] use differential testing to compare analyzer outputs and detect precision issues. Soundness-Bench [24] is related in motivation but differs in scope and granularity. SoundnessBench evaluates end-to-end verifier correctness on crafted benchmark instances with counterexamples, asking whether a verifier returns the correct final verdict. SoundnessBench stresses final-verdict correctness, while our method provides diagnostic evidence about where and why concrete–abstract inconsistency arises inside a verifier. Our work follows this line of research, but targets neural network robustness verifiers by enforcing two forms of concrete–abstract consistency: (1) refuting incorrect certification claims via concrete counterexamples, and (2) localizing incorrectness when concrete activations fall outside their corresponding abstract bounds.

**Robustness verification methods.** Neural network robustness verification spans complete SMT/MILP-style reasoning for piecewise-linear networks (e.g., *Reluplex* and Marabou) [11, 12], scalable abstract-interpretation-style bound propagation (e.g., *Deep-Poly/DeepZ*) [19], and optimization-based linear relaxation frameworks such as LiRPA and auto_LiRPA [23], with branch-and-bound hybrids such as $\alpha, \beta$-Crown [21]. Standardized evaluation efforts like VNN-COMP and specification formats such as VNN-LIB enable broad comparisons across tools [1, 2, 5, 13], and frontends such as DNNV further decouple parsing from backends [16]. Unlike these works, we do not propose a new verification algorithm; we focus on detecting the unsoundness of verifier implementations.

## 5 Conclusion

We present an unsoundness detection framework for robustness verifier. Our framework combines two complementary checks. CBR refutes false certifications by searching for concrete counterexamples in the input region. BBL checks a necessary internal invariant (i.e., concrete activations lie within their abstract bounds) and localizes soundness-relevant faults to specific layers or operators, even when CBR is inconclusive due to limited sampling or high-dimensional inputs. We evaluate NN verifiers across three abstract domains. Across all the soundness-violating instances, our framework detects 72% of injected soundness violations. We believe it provides a promising approach for testing the soundness of a range of neural network verification algorithms and for helping developers debug and localize soundness issues in their implementations.

## Acknowledgments

We thank the anonymous reviewers for their valuable feedback and suggestions. We also acknowledge the use of the generative AI tool ChatGPT for language polishing and grammar checks.

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
