# OpenReview forum: "Detecting Unsoundness in Neural Network Verifiers via Concrete–Abstract Consistency"
_ACM.org/AIWare/2026/Conference — AIware 2026_

### Official Review · Reviewer_SvsZ · 2026-03-02

**Rating:** 3
**Confidence:** 4

**Review:**

Overall, I found the paper to be well-written, and quite interesting. Testing neural network verifiers appears to be an interesting and possibly overlooked research direction. Furthermore, the methodology appears to be novel and sensible, as attempting to find counterexamples and use boundary invariants in neural network layers can indeed be helpful in order to detect unsoundness in neural network verifiers. The algorithms of the directions followed are clearly presented and well-articulated. Finally, the evaluation process and the selected experiment set appears to be comprehensive, and the results are presented in a clear and coherent manner and are structured well.

Below are some additional comments regarding this work.

**Novelty:**
Upon reviewing the literature, I found that SoundnessBench [1] also employs counterexamples to evaluate neural network verifiers. While the authors include SoundnessBench in the paper, and it appears to follow a different methodology (utilizing neural network training to generate hidden counterexamples) it is important for the authors to clarify how their approach differs from [1]. I believe the current work presents a novel contribution, but this novelty should be more clearly articulated in relation to [1] and the broader literature.

**Methodology:**
Overall, I found the methodology to be solid and well-explained. Some basic comments:
- For your counterexample approach, you mention you use a sampling budget of 20 concrete inputs per NN for that purpose. How did you come up with this number and why? Would you expect different results using a larger or a smaller sampling budget? Please clarify this.
- Regarding evaluation metrics, while you mention them in detail, it is not clear how you chose them for evaluating your experiments. Please further elaborate in this direction.
- Are there any additional mutation operators that would be interesting to further explore as future work?

**Related Work:**
The related work section includes the most relevant publications but requires further improvement:
- The subsection related to analyzer soundness is overly condensed, mainly listing references without sufficient discussion. It should better explain the key ideas of the cited works and clarify their relevance to the current contribution.
- The robustness verification subsection mentions existing tools but does not clearly explain how this work differs from them. The authors should elaborate on the specific distinctions and underlying considerations behind their approach.
- Overall, the authors should explicitly discuss whether any prior works share methodological similarities with their approach to better position their contribution within the literature.
- I recommend that you further search the literature and include work related to mutation testing of neural networks, such as DeepCrime [2] and DeepMutation [3]. While these works are more broadly related to your topic, including them would benefit readers by providing additional context on this relevant direction.

**Minor Issues:**
- The formula and some details on part (3) of Figure 1 are difficult to read. Please revise and refine it.
- Table 1 can be condensed to save space.

[1] https://openreview.net/forum?id=UuYYldVLH3

[2] https://dl.acm.org/doi/10.1145/3460319.3464825

[3] https://www.computer.org/csdl/proceedings-article/issre/2018/832100a100/17D45W1Oa1G

**Summary:**

The authors propose a methodology in order to detect unsoundness in verifiers of neural networks. To do so, they apply a set of strategies, such as attempting to find counterexamples in a certification to refute it, or check if specific concrete neural network layer activations fall within abstract boundary invariants set by the verifier. To this direction, the authors implemented six mutation operators in a tool called NetFactory capable of implementing mutated NNs based on the strategies described. Using NetFactory, the authors conducted experiments on 450 soundness-violating instances from 30 neural networks across three different domains. Overall, NetFactory was able to detect 73% of the injected soundness violations across the experiment set.

---

> ### Author Response · Authors · 2026-03-17
>
> ## Response to Reviewer SvsZ
>
> We thank the reviewer for the constructive feedback. We address each comment below.
>
> ### C1-SvsZ: Relation to SoundnessBench
>
> We appreciate this suggestion. The current manuscript cites SoundnessBench, but the distinction can indeed be made clearer. SoundnessBench evaluates end-to-end verifier correctness on crafted benchmark instances with known counterexamples; it asks whether a verifier returns the correct final verdict. Our framework instead targets implementation-level unsoundness by auditing concrete-abstract consistency and localizing faults inside the verifier. In particular, CBR detects output-level contradictions between certification and concrete behavior, while BBL checks per-layer containment and localizes faults to specific layers or operators. We will sharpen this distinction in the revision and position the two approaches more explicitly as complementary rather than competing.
>
> ### C2-SvsZ: Clarification on sampling budget
>
> The budget of 20 was selected as a practical default based on preliminary experiments. RQ6 (Table 9) already reports runtime across budgets {5,10,20,50}. We ran a controlled sub-evaluation (30 networks, 5 mutations, 150 instances in the *interval* domain, identical seeds so that only the sampling count varies). The resulting detection rates are: CBR remains near 15-16% while runtime scales linearly, BBL remains effectively constant (74.0% at budgets 5-20, 75.3% at budget 50). In this controlled sub-evaluation, budget =20 is therefore a practical default: increasing to 50 yields only a small gain (+2pp) while more than doubling runtime. We will add this analysis alongside the existing RQ6 runtime results.
>
> ### C3-SvsZ: Clarification on evaluation metrics
>
> The evaluation metrics are already defined in Section 3.1 and are aligned with the research questions: detection rate (RQ1), localization accuracy (RQ3), operator coverage (RQ4), and overhead (RQ6). Their intent is to measure, respectively, whether soundness faults are found, whether BBL provides useful debugging guidance, whether NetFactory exercises the supported operator space broadly enough, and what practical cost the checks introduce. We will make this metric rationale more explicit in the revision.
>
>
> ### C4-SvsZ: Future extensions of the mutation set
> This is a useful suggestion. Our current mutation set focuses on bound-level soundness faults because these support controlled ground truth and precise localization. We agree that broader mutation families would further strengthen the methodology. Promising future directions include semantic mutations (e.g., operator substitutions), topology-level mutations, precision-related mutations, and solver-/encoding-level mutations. We will add a short discussion of these directions in the revision.
>
> ### C5-SvsZ: Planned strengthening of related work
> We appreciate this suggestion and will strengthen the related-work section in three ways. First, we will expand the analyzer-soundness subsection to better explain the key ideas of Bugariu [3], Klinger et al. [14], and He et al. [8], and clarify how our concrete-abstract consistency approach relates to them. Second, we will more clearly separate verification algorithms (e.g., Reluplex, DeepPoly, and $\alpha,\beta$-CROWN) from our testing framework for verifier implementations. Third, we will add DeepCrime and DeepMutation as broader mutation-testing context, while noting that these works mutate DNN models whereas we mutate verifier implementations.
>
> ### C6-SvsZ: Minor presentation revisions
>
> We agree that part (3) of Figure 1 is difficult to read and that Table 1 can be condensed. We will revise both in the next version.

---

> > ### Comment · Reviewer_SvsZ · 2026-03-18
> >
> > Thank you for addressing my comments. I believe this is a valuable contribution, and, given that you address all aspects as you mention, I will maintain my assessment for paper acceptance, and increase my confidence.

---

### Official Review · Reviewer_4LVT · 2026-03-08

**Rating:** 3
**Confidence:** 3

**Review:**

Strengths:
- The paper focuses on detecting unsoundness in neural network verifiers, which is a critical and underexplored problem.
- The experimental results are promising, demonstrating a high detection rate for injected soundness violations.
- The authors provide an open-source implementation.

Weaknesses:
- The evaluation scope is relatively narrow, which limits the generalizability of the findings.
- The random sampling strategy used in CBR can be inefficient, especially in high-dimensional input spaces.
- The scalability of the proposed framework remains unclear.

Comments:

1. The evaluation appears limited in terms of generalizability. In particular, the framework is evaluated using self-implemented verifiers with manually injected mutations. Although this limitation is acknowledged in the paper, it still raises concerns about how representative the evaluation is of real-world scenarios. For example, how do the designed mutations reflect the types of unsoundness that occur in practice? Are these mutations derived from empirical observations of real-world verifier failures, or are they mainly synthetic constructs? Moreover, can they be applied to real-world verifiers without substantial modification? The authors should clarify these points to make the paper's claims more convincing.

2. The evaluation is restricted to feed-forward networks and does not include widely used architectures such as Transformers, RNNs/LSTMs, or graph neural networks. This raises concerns about the applicability and scalability of the proposed framework to other architectures. Extending the evaluation to these models would help demonstrate the broader usefulness of the approach.

3. The random sampling strategy used in CBR may be inefficient. Since the counterexample search relies on random sampling within a fixed budget, it may suffer from very low hit rates in high-dimensional input spaces and could easily miss violations. It would be helpful for the authors to discuss or explore more guided sampling strategies, such as coverage-guided or gradient-based input generation, to improve the effectiveness of CBR. In addition, the choice of sampling budgets (5, 10, 20, 50) appears somewhat arbitrary. An analysis of how the sampling budget affects both detection rate and runtime would help justify these settings.

**Summary:**

This paper presents an unsoundness detection framework for robustness verifier implementations. The framework combines two concrete–abstract consistency checks, including Counterexample-Based Refutation and Bounds-Based Localization. To reduce representation drift, the authors use specification-embedded models that wrap the core neural network with input and output specifications as two additional layers. They further develop an operator-aware NN generator that produces diverse models spanning a wide range of layer types, parameters, and architectures. They conducted evaluation on neural network verifiers across three abstract domains using six mutation operators. The experimental results indicate that the proposed framework detects 73% of injected soundness violations across 450 soundness-violating instances, showing that combining coarse refutation with fine-grained invariant checking provides practical assurance for robustness verifiers.

---

> ### Author Response · Authors · 2026-03-17
>
> ## Response to Reviewer 4LVT
>
> We thank the reviewer for the constructive feedback. We address each comment below.
>
> ### C1-4LVT: Mutation design and evaluation scope
>
> Our evaluation uses controlled injected faults to obtain measurable ground truth for both detection and localization, as already stated in Section 3.1 and acknowledged as a limitation in Section 3.9. The current mutation set is intended to approximate common verifier-side failure modes rather than arbitrary perturbations (Table 3). We agree that evaluation on independently developed third-party verifiers would further strengthen the paper, and we will make this limitation more prominent in the revision.
>
> ### C2-4LVT: Clarification on network architectures
>
> We agree that the current evaluation covers only MLP and CNN architectures. This scope limitation is already stated in Section 3.1, which defines the current evaluation over MLP/CNN2D families including plain, residual, and staged variants, which align with the dominant architectural families used in current verification benchmarks (e.g., ACAS Xu and MNIST-style classifiers). Section 3.9 explicitly notes that the present results do not yet cover architectures such as Transformers or RNNs. We will clarify more explicitly in the revision.
>
> ### C3-4LVT: CBR sampling and budget choice
>
> As discussed in the Abstract and in Section 3.2 and 3.3, the low CBR detection rate in RQ1 is primarily an observability issue rather than a sampling-density issue: many mutations affect intermediate bounds without producing output-spec violations, making them structurally invisible to CBR. This is precisely why CBR is paired with BBL. This is the pattern reported in Table 4: CBR-only detects 14%, BBL-only 71%, and the combined workflow 73%. Section 3.7 (RQ6) already reports runtime across budgets {5,10,20,50}; we agree that the rationale for choosing 20 as the default budget can be stated more clearly, and we will make this explicit in the revision.
>
> ### C4-4LVT: Clarification on current scalability
>
> Section 3.7 (RQ6) already evaluates three model sizes (approximately 1K, 33K, and 297K parameters). Across this tested range, BBL remains near-constant (0.11 - 0.15 ms), CBR remains below 2 ms at budget 20, and the combined overhead remains small relative to the verifier's analysis runtime. We agree that this does not yet establish scalability to substantially larger models, and we will state this limitation more explicitly in the revision.

---

> > ### Comment · Reviewer_4LVT · 2026-03-19
> >
> > Thank you for the response. The authors have addressed my concerns and committed to including further details in the revision. My overall rating remains unchanged.

---

### Official Review · Reviewer_rEQh · 2026-03-08

**Rating:** 3
**Confidence:** 4

**Review:**

**Strengths**

* Addresses the important problem of soundness in neural network verification tools
* Clear framework combining counterexample refutation and bound consistency checks
* Layer-level fault localization useful for debugging verifier implementations
* Operator-aware network generator to exercise different verifier operators
* Empirical study demonstrating the detection of many injected unsoundness violations

**Weaknesses**

* Evaluation relies on injected faults rather than real-world verifier bugs
* Limited diversity of verification implementations evaluated
* Synthetic networks may not represent realistic verification workloads

**Detailed Review Comments**

The paper investigates the problem of detecting unsoundness in neural network robustness verifiers. This is an important and timely problem because verification tools are increasingly used to provide assurance for safety-critical systems, and implementation-level bugs in these tools can lead to incorrect certification results. The proposed framework introduces two complementary mechanisms: Counterexample-Based Refutation (CBR), which attempts to find concrete inputs that invalidate a verifier’s claim, and Bounds-Based Localization (BBL), which checks whether concrete activations violate the abstract bounds computed by the verifier. The separation between these two mechanisms is conceptually clear, with CBR providing coarse-grained detection and BBL enabling finer-grained fault localization within network layers. The framework also includes NetFactory, an operator-aware neural network generator designed to exercise different operator implementations within verification tools.

However, several limitations affect the strength of the empirical evaluation.

1. Evaluation relies entirely on injected faults: The study introduces mutation operators to create artificial soundness bugs in verifier implementations. While mutation testing provides a controlled way to evaluate detection capability, the paper does not demonstrate the framework on real-world verifier bugs or historical defects. As a result, it remains unclear whether the proposed approach can detect naturally occurring unsoundness issues in practice.

2. Limited diversity of verification implementations: Although the experiments include three abstract domains (interval, HybridZ, and dual), the implementations share the same interval arithmetic component. Consequently, the domains exhibit identical behaviour in the experiments, limiting the ability to demonstrate the framework’s applicability across different verification algorithms.

3. Limited effectiveness of counterexample-based detection: The results show that Counterexample-Based Refutation detects only a small fraction of injected violations. Most detections come from the bounds-based localization mechanism. This suggests that the effectiveness of CBR may depend strongly on sampling strategies and input dimensionality, and the paper could further discuss when this mechanism is expected to succeed.

4. Synthetic network generation may limit external validity: The evaluation relies on neural networks generated by NetFactory rather than real benchmark models. Although the generator can produce different architectures, it is unclear whether these networks resemble models used in existing verification benchmarks. Evaluating the framework on standard benchmarks would strengthen the generalizability of the results.

5. Detection coverage remains incomplete: The framework detects approximately 73% of injected soundness violations, leaving a significant portion undetected. The paper would benefit from a deeper analysis of these missed cases, including whether they correspond to particular operators, mutations, or limitations of the detection mechanisms.

**Summary:**

The paper presents a framework for detecting unsoundness in neural network robustness verifiers. The approach is based on two complementary validation mechanisms. Counterexample-Based Refutation (CBR) attempts to identify concrete inputs that invalidate a verifier’s certification claim by demonstrating that the network violates the claimed property. Bounds-Based Localization (BBL) checks whether the concrete activations produced during execution remain within the abstract bounds computed by the verifier, which helps identify potential implementation errors and localize faults within the verification process. To systematically exercise different verifier behaviours, the framework introduces NetFactory, an operator-aware neural network generator that produces diverse neural network architectures designed to trigger different transfer functions within verification systems. The evaluation injects artificial soundness bugs using mutation operators across three abstract domains (interval, HybridZ, and dual) and assesses whether the proposed techniques can detect these violations. Experimental results show that the combined use of CBR and BBL detects 73% of the injected unsoundness violations while also providing useful information for fault localization.

---

> ### Author Response · Authors · 2026-03-17
>
> ## Response to Reviewer rEQh
>
> We thank the reviewer for the constructive feedback. We address each comment below.
>
> ### C1-rEQh: Clarification on controlled fault injection
>
> This choice is methodological: evaluating detection and localization requires known fault types and injection locations. Our mutations therefore target verifier-side failure modes rather than arbitrary perturbations (Table 3), with Section 3.8 illustrating two representative cases. We acknowledged this in Section 3.1, where we explain that mutation injection is used to obtain controlled ground truth for both detection and localization, and in Section 3.9 (Limitations) we explicitly state that the current study evaluates our own implementations with injected soundness bugs rather than independently developed third-party verifiers.
>
> ### C2-rEQh: Diversity of verification implementations
>
> As reported in Section 3.6, all three domains compute equivalent bounds on the tested networks, so BBL reports identical containment outcomes in Table 8. We have already acknowledged this in Section 3.6 that *HybridZ* and *dual* currently share the same interval-arithmetic core, and therefore exhibit identical behavior on the present experiments. Accordingly, RQ5 is intended as a baseline validation of the comparison harness, rather than as evidence of strong cross-domain differentiation.
>
> ### C3-rEQh: Effectiveness of counterexample-based detection
>
> As discussed in the Abstract and in Section 3.2 and 3.3, CBR is designed as a lightweight, output-level complement to BBL, not as a standalone detector. Its effectiveness depends on whether an internal fault propagates to an output-spec violation on sampled executions. When a mutation affects only intermediate bounds without changing the output verdict, CBR is structurally unable to detect it regardless of sampling budget which is exactly why BBL checks the internal containment invariant layer by layer. This is the pattern reported in Table 4: CBR-only 14%, BBL-only 71%, combined 73%. RQ2 further shows that CBR performs well when violations are output-visible, confirming that the bottleneck is observability rather than sampling density.
>
> ### C4-rEQh: Synthetic network generation
>
> This limitation is acknowledged in Section 3.1 and 3.9. In the current study, the generated networks are not toy cases. As shown in Table 2, NetFactory covers MLP and CNN2D families, including plain, residual, and staged variants, which align with the dominant architectural families used in current verification benchmarks (e.g., ACAS Xu and MNIST-style classifiers). We agree that structural similarity does not replace direct benchmark evaluation and will position this as a future step.
>
> ### C5-rEQh: Clarification on undetected cases
> We agree that the remaining undetected cases deserve a clearer explanation. The main reason is not a uniform blind spot, but observability: some mutations affect intermediate abstract bounds without changing the sampled concrete behavior enough to surface as either an output-spec violation (for CBR) or a containment violation (for BBL). This is most likely when the injected perturbation is small and remains latent on the sampled execution. We already discuss this mechanism qualitatively in Section 3.2 and 3.3, and we will make this explanation more explicit in the revision.

---

> > ### Comment · Reviewer_rEQh · 2026-03-18
> >
> > Thank you for the clear response and your efforts in addressing the comments. The clarification of CBR vs. BBL, especially the observability limitation, addresses my main concern. The justification for mutation-based evaluation is reasonable, though the lack of real-world verifier bugs and independent implementations still affects the external validity.
> >
> > Overall, the response improves clarity and my understanding; I maintain my assessment with increased confidence.